# New Horizons in Early Dementia Diagnosis: Can Cerebellar Stimulation Untangle the Knot?

**DOI:** 10.3390/jcm8091470

**Published:** 2019-09-16

**Authors:** Antonino Naro, Angela Marra, Luana Billeri, Simona Portaro, Rosaria De Luca, Giuseppa Maresca, Gianluca La Rosa, Paola Lauria, Placido Bramanti, Rocco Salvatore Calabrò

**Affiliations:** IRCCS Centro Neurolesi Bonino Pulejo, via Palermo, S.S. 113, C.da Casazza, 98124 Messina, Italy; g.naro11@alice.it (A.N.); angela.marra@irccsme.it (A.M.); luana.billeri@irccsme.it (L.B.); simona.portaro@irccsme.it (S.P.); rosaria.deluca@irccsme.it (R.D.L.); giuseppa.maresca@irccsme.it (G.M.); gianluca.larosa@irccsme.it (G.L.R.); paola.lauria@irccsme.it (P.L.); dino.brama@gmail.com (P.B.)

**Keywords:** dementia, Mild Cognitive impairment, brain connectivity, cerebellum, cerebellum-cerebrum connectivity (CCC)

## Abstract

Differentiating Mild Cognitive Impairment (MCI) from dementia and estimating the risk of MCI-to-dementia conversion (MDC) are challenging tasks. Thus, objective tools are mandatory to get early diagnosis and prognosis. About that, there is a growing interest on the role of cerebellum-cerebrum connectivity (CCC). The aim of this study was to differentiate patients with an early diagnosis of dementia and MCI depending on the effects of a transcranial magnetic stimulation protocol (intermittent theta-burst stimulation -iTBS) delivered on the cerebellum able to modify cortico-cortical connectivity. Indeed, the risk of MDC is related to the response to iTBS, being higher in non-responder individuals. All patients with MCI, but eight (labelled as MCI-), showed preserved iTBS aftereffect. Contrariwise, none of the patients with dementia showed iTBS aftereffects. None of the patients showed EEG aftereffects following a sham TBS protocol. Five among the MCI- patients converted to dementia at 6-month follow-up. Our data suggest that cerebellar stimulation by means of iTBS may support the differential diagnosis between MCI and dementia and potentially identify the individuals with MCI who may be at risk of MDC. These findings may help clinicians to adopt a better prevention/follow-up strategy in such patients.

## 1. Introduction

Dementia is a clinical syndrome encompassing progressive cognitive impairment (involving, among other, memory, language, and behavioral functions) that leads to difficulties in activities of daily living up to total dependence. Alzheimer’s disease, vascular dementia, mixed dementia, and dementia with Lewy bodies are the most common types [1]. Dementia has become a worldwide concern due to the global population aging, the considerable burden on demented individuals and their families, and healthcare and social costs [1]. Neuropsychological assessment has a fundamental role in the diagnosis of dementia and Mild Cognitive Impairment (MCI), a condition in which patients complain of difficulties in at least one cognitive domain among memory, reasoning, planning or problem solving, attention, language, or visual depth perception, without major problems with everyday living. Noteworthy, is that this condition often foreruns dementia (namely, MCI-to-dementia conversion, MDC) [2,3,4,5]. In addition, there is a growing need of identifying paraclinical approaches to early diagnose dementia and identify the patients who may be at risk of MDC, so to plan a patient-tailored pharmacological and rehabilitative management.

Some EEG features (including EEG power density, functional coupling, and spectral coherence) have been tested for their usefulness in differentiating dementia and MCI at the prodromal stage [4,5]. It has been suggested that there is a correlation among neurodegeneration, the amount of cognitive impairment, and the alterations of brain oscillations in patients with dementia and MCI [6,7,8]. Indeed, MCI and dementia lead to a different degree of connectivity deterioration among the brain areas involved in cognitive functions, because of deposition of amyloid-β plaques and neurofibrillary tangles, neuronal loss, vascular damage, and various other degenerative processes [9,10,11,12,13,14,15,16,17,18]. Thus, both dementia and MCI can be regarded as a disconnection syndrome [9,10,11,12,13,14,15,16,17,18], further corroborating the issue that MCI can evolve into dementia [9,10,11,12,13,14,15,16,17,18,19,20,21]. However, not all the patients with MCI evolve into dementia. This may depend, among other things, on the heterogeneity of the MCI syndrome (e.g., amnesic versus non-amnesic MCI) [4]. However, the neurophysiological basis on which a patient with MCI converts to dementia remains to be clearly elucidated [4]. Further, cortical connectivity deterioration in resting condition does not always linearly correlate with cognitive decline [6,7,8,22,23,24,25,26]. In fact, some individuals with early MCI and dementia may disclose abnormal connectivity and oscillatory patterns only under specific conditions, e.g., when performing a task. Thus, the risk of MDC is not easily foreseeable, and the differential diagnosis between dementia and MCI at the prodromal stage remains challenging [2,3]. 

New objective markers are consequently necessary to early differentiate MCI and dementia and identify the patients at risk of MDC. Given that the paraclinical assessment in resting condition is not necessarily either conclusive or informative, perturbing brain connectivity, e.g., by using non-invasive neurostimulation, may allow the clinician to identify changes in connectivity that could be undetectable during the resting condition, or possibly misinterpreted during the administration of cognitive tasks [3,27,28,29,30,31]. The cerebellum has been growingly emphasized among the potential target of non-invasive neurostimulation to induce changes in cortico-cortical connectivity [32,33,34,35,36,37,38,39,40,41,42,43,44,45,46]. This is not surprising, if we take into account the growing evidence on complex and multiple connections of the cerebellum with the cerebrum to modulate cortical connectivity within the networks that are critical to orchestrate cognitive, emotional, and sensorimotor processes (including the default mode network and frontoparietal networks) [20,33,34,35,36,37,38,39,40,41,42,43,44,45,46,47].

To date, a few studies employing fMRI and EEG/MEG explored cerebellum-cerebrum connectivity (CCC) to provide an early dementia/MCI diagnosis [14,19,40,48,49,50,51,52,53]. These studies have altogether reported a more severe bundled detrimental CCC in patients with dementia compared to those with MCI [13,19,33,40,48,49,54,55]. However, the functional consequences of the deteriorated CCC in terms of cognitive performance has been understood only partially, given that people with early dementia and MCI may not necessarily show a different degree of brain and CCC impairment at rest [33]. Therefore, the advantage of perturbing cortical connectivity using non-invasive neuromodulation of CCC stems from the fact that patients with dementia could show a worse response of cortical connectivity to non-invasive neuromodulation than patients with MCI. This may depend on a greater impairment of the cerebellar output onto frontoparietal networks.

A useful approach to assess in vivo the deterioration of CCC at rest, may consist in the analysis of the effects of non-invasive cerebellar stimulation on brain network connectivity measured using EEG. It is indeed well-known that cerebellar transcranial magnetic and electric stimulation, including theta-burst stimulation (TBS), largely affect brain connectivity [3,21,27,28,29,31,56,57,58,59,60,61,62,63,64,65]. We hypothesize that the magnitude of the aftereffects of non-invasive cerebellar stimulation, with regard to the intermittent TBS (iTBS), on brain network connectivity (measured by means of the EEG) may be used to differentiate patients with early dementia and MCI. Indeed, patients with dementia could show a worse response of cortical connectivity to iTBS than patients with MCI could do. We employed iTBS as a conditioning paradigm, given that this has been shown to induce changes in cortical excitability within sensorimotor and non-motor areas [3,21,27,28,29,31,56,57,58,59,60,61,62,63,64,65,66,67,68]. iTBS aftereffects presumably occur by a modulation of cerebello-thalamo-cortical pathways (encompassing the dentate nucleus and the ventrolateral motor thalamus in a dysinaptic excitatory pathway onto the mechanisms of intracortical inhibition and of cholinergic activity within M1 and different non-motor interconnected regions) [62,69,70]. We stimulated both the cerebellar hemispheres because cerebral-cerebellar connectivity has a complex nature and the target cerebellar hemisphere to cerebral connectivity is still not well defined. Given that cerebellar iTBS may reveal a dysfunctional brain connectivity, it is furthermore hypothesizable that the patients with MCI who do not respond to TBS (labelled as MCI-, similarly to the patients with dementia), may be at a higher risk of MDC than those who normally respond to iTBS (labelled as MCI+).

## 2. Materials and Methods

### 2.1. Subjects

We enrolled 25 patients with MCI (both amnestic, aMCI, and non-amnestic, naMCI) and 24 with probable dementia (probable Alzheimer’s, AD, and vascular dementia, VaD) between March 2018 and January 2019. The clinical diagnosis was achieved according to the available international diagnostic guidelines [71,72,73,74,75,76]. The exclusion criteria were: (i) evidence of other neurological or psychiatric diseases leading to cognitive impairment; (ii) uncontrolled or complicated systemic diseases or traumatic brain injuries; (iii) epileptic history or electroencephalographic (EEG) epileptiform activity; (iv) contraindication to TMS (including epileptic history, electro/mechanical devices within the head and neck, pacemaker, aneurysm clips, neurostimulator, and brain/subdural electrodes); and (v) use of any psychoactive medication other than antidepressants and donepezil. Clinical-demographic characteristics are summarized in Table 1. The Local Ethics Committee approved the present study (ID: IRCCSME 29/18). All participants gave their written informed consent.

### 2.2. Experimental Procedure

EEG functional connectivity was evaluated in each participant at baseline (T_0_). Then, an iTBS paradigm was applied on the lateral cerebellum. iTBS aftereffects onto EEG functional connectivity were thus recorded after 10 min from protocol end (T1) (Figure 1). This time was chosen because the ability to modulate plasticity of iTBS starts already from 1–3 min and lasts up to ~one hour after protocol application, so to be sure to detect real and not casual iTBS aftereffects. Further, the patients were subjected to the same experimental paradigm but using a sham-coil iTBS protocol in a separate session (at least one week apart). The order of the two experimental paradigms was random. Last, patients were clinically assessed six months after iTBS administration.

TMS was carried out using a figure of eight coil (90 mm wing diameter) wired to a MagStim stimulator (MagStim Co., Whitland, UK) for TMS basic measures (resting, RMT, and active motor threshold, AMT; motor evoked potential, MEP) and a Rapid^2^ MagStim for iTBS provision.

First, we determined the so-called hotspot, i.e., the point over M1 where the coil with the handle pointing backwards and laterally induced a posterior-to-anterior directed electrical current in the brain that consistently evoked the largest MEP with the steepest initial slope in the relaxed first dorsal interosseous muscle (FDI) of the right hand. Then, we determined the RMT and AMT of M1 by using single-pulse TMS stimuli with the coil positioned over the M1 hotspot. These measures were used to set up iTBS protocol. Specifically, RMT was defined as the minimum intensity that evoked a peak-to-peak MEP of 50 µV in at least 5 out of 10 consecutive trials in the relaxed FDI muscle. AMT was defined as the minimum intensity that elicited a reproducible MEP of at least 200 μV in the tonically contracting FDI muscle in at least 5 out of 10 consecutive trials. Participants maintained a force level of approximately 10–15% of maximum force during measurements of the active MT.

To carry on iTBS, the abovementioned figure of eight coil was centered over the lateral cerebellum (3 cm lateral and 1 cm inferior to the inion), positioned tangentially to the scalp, with the handle pointing superiorly [52]. The order of lateral cerebellum to be stimulated was random, with a 2 min rest between hemispheres. The exact coil position was marked by an inking pen to ensure the same coil positioning throughout the experiment. A mechanic arm held the coil, which position was continuously monitored throughout the experiment. The magnetic stimulus had a biphasic waveform with a pulse width of about 300μs. During the first phase of the stimulus, the current in the center of the coil flowed toward the handle. iTBS consisted of a 2 s train of TBS (three-pulse bursts at 50 Hz) repeated 20 times, every 10 s for a total of 190 s (600 pulses) [52]. TMS intensity was set at 80% AMT (tested over the motor cortex of the left hemisphere).

### 2.3. EEG Recording and Analysis

EEG was recorded for at least 10 min while the subjects were sitting in a comfortable recliner chair in resting state in a quiet and mild-lighted room, with the eyes closed, before (T0) and 10 min after the end of iTBS administration (T1). To the purpose of an easy clinical application of the paradigm, we used a standard headset with 19 Ag-AgCl disk electrodes (i.e., Fp1/2, F7/8, F3/4, Fz, T3/4, T5/6, C3/4, Cz, P3/4, Pz, O1/2) placed according to 10-20-International System. The ground was put on the forehead, the reference on both mastoids. Eye movements and blinks were detected by an EOG (two additional electrodes above the right and below the left eye). Data were acquired using a Brain-Quick System (Micromed; Mogliano Veneto, Italy), sampled at 512 Hz, and filtered at a 0.3/70 Hz band-width (with 50 Hz notch). Skin-electrode impedance was always below 5 kOhm. Data were stored on a personal computer for offline analysis through a free license of EEGLAB toolbox. EEG artifacts were identified and removed by using visual inspection and ICA. Artifact-free EEG was thus segmented into 2 s epochs (as at least 2-s data of continuous artifact-free EEG recordings as one epoch are required for low-resolution electrical tomography –LORETA- analyses), thus obtaining 300 ± 50 epochs at each time T. 

Standardized LORETA (sLORETA) was thus used to analyze the cortical distribution of current source density. Computations were made using the Montreal Neurological Institute template (MNI152) template restricted to cortical gray matter (6239 voxels at 5 mm spatial resolution) according to the probabilistic Talairach atlas [77,78,79]. Anatomical labels as Brodmann areas were reported using MNI space with correction to Talairach space [80,81]. sLORETA was calculated from 0.5 Hz to 30 Hz in six frequency bands: delta (2–4 Hz), theta (4–8 Hz), alpha1 (8–10 Hz), alpha2 (10–13 Hz), beta1 (13–20 Hz), beta2 (20–30 Hz).

Functional connectivity was analyzed between cortical regions of interest (ROIs) determined according to a voxel-wise approach. Because there are still no clear rules for determining ROIs, in this study sLORETA defined the MNI coordinates of the cortical voxels underlying the 19 electrode sites to create the ROIs, in keeping with previous works [82,83]. These so-obtained cortical areas are also well documented in other low-resolution tomography approaches [84]. Then, we determined the centroid voxel per region (i.e., one voxel per region that has the minimum average Euclidean distance to the rest of voxels in that brain region). Last, we applied principal component analysis (PCA) to each of the centroid voxels separately. We thus extracted the orientation of the source-dipole dynamics within each ROI. The first PCA projection from a single centroid-voxel time-series was considered for the data analysis.

One may be concerned on the fact that the use of a canonical template and canonical electrode positions for the calculation of the forward model for connectivity in brain may lead to weak connectivity data. However, it has been shown that there is no systematic bias or inconsistency between template and native MRI co-registration of brain electric signals for the analysis of functional connectivity, even in small samples [85].

The data analysis aimed at measuring changes in the intrinsic connectivity in the source-reconstructed EEG obtained above as a result of iTBS neuromodulation (otherwise a generic intervention, like motor task or motor learning). Specifically, we investigated how a single session of iTBS changed cortico-cortical coherence (Coh) during resting-state following iTBS application. To this end, we estimated ROI-wise Coh in each frequency range and for post-pre iTBS differences in resting state EEG [51,86,87,88,89]. Coh quantifies the frequency and amplitude of the synchronicity of neuronal patterns of oscillation. Specifically, it measures the synchrony between signals from different electrodes at each Fast Fourier transform (FFT) frequency bin. FFT for spectral decomposition was carried out on 5-s epochs with a 50% overlap and averaged for each frequency band. Thus, the absolute values of Coh (i.e., coherence or magnitude coherence) were calculated as the magnitude of the average cross-spectrum (The average cross-spectrum is obtained by computing the product of the FFT of two signals. It indicates how much energy in the frequency band is common to both the signals) between the two signals at a certain frequency range divided by the square root of the power spectral density (PSD) (PSD is obtained by computing the power spectra of each signal using FFT. It describes where the average power is distributed as a function of frequency (expressed in µV2/Hz)) of both the signals at the frequency range [51,86,87,88]. 

The imaginary part of coherency (iCoh; sometimes named imaginary part of coherence) [51,86,87,88,89] was then extracted between all voxel combinations and each frequency range for post-pre iTBS condition for all participants. Thus, we obtained a multidimensional functional connectivity array (i.e., time × frequency × voxel combinations). Last, iCoh was corrected by its real counterpart (real part of coherency, sometimes named real part of coherence), thus obtaining the so-called corrected imaginary part of coherency/coherence. iCoh ignores the spurious instantaneous interactions due to volume conduction, thus taking into account only the non-instantaneous interactions.

### 2.4. Statistical Analysis

Multivariate statistical analysis was used to compare the so-obtained post-pre iTBS functional connectivity arrays by employing the Partial Least Squares (PLS). This data-driven approach easily identifies the changes in intrinsic connectivity induced by any intervention (e.g., brain stimulation techniques, including iTBS, and motor learning). The simple contrast between pre- and post-iTBS aftereffects on iCoh (obtained by subtracting post-iTBS from pre-iTBS iCoh values and then averaged across subjects) represented the independent variable. The pre-iTBS iCoh represented the dependent variable [51,86,87,88,89]. 

We thus elaborated the group-level PLS by averaging iCoh for post-pre iTBS contrast, thus yielding iCoh values as a function of group × frequency × voxel. Then, the contrasted (i.e., post-pre iTBS) iCoh values were averaged across subjects, yielding grand-average group-level functional connectivity array as a function of frequency × voxel combinations. Therefore, we generated surrogate data using permutation testing [90,91]. We randomly permuted the connectivity arrays for pre- and post-iTBS proceeding as for the original data. This analysis was performed for 10000 realizations. PLS components were thus compared with surrogate data and considered statistically significant if their eigenvalue exceeded 95% of the corresponding eigenvalues of the surrogate distribution (*p* < 0.05). 

As post-hoc analysis, we compared the average iCoh values between pre- and post-iTBS to identify the direction of the changes. About that, we employed paired *t*-tests to assess whether the EEG phase difference and the Hurst component of iCoh values were significantly different between pre- and post-iTBS [51,86,87,88,89,92]. EEG phase difference refers to the delay for each of the frequency components of a signal, calculated as phase/2π*f* (*f* is the frequency). In our case, it estimated the phase direction of functional connectivity among pairs of ROIs as influenced by cerebellar stimulation [92]. Hurst component allows distinguishing random from non-random systems and identifying the persistence of trend in a signal [93,94]. In other words, it provides a unitary measure of the long-term nonlinearity of a signal, that is, an estimate of the tendency of a time series either to regress strongly to the mean or to cluster in a direction. Hence, we obtain a power law decay/increase, whose exponent is the Hurst component. Therefore, a Hurst component increase or decrease means that the values along time will also tend to be high or low, respectively, into the future, suggesting a trend to connectivity increase or decrease, respectively.

The correlations between clinical (Clinical Dementia Rating scale) and electrophysiological measures (overall iTBS aftereffects) were tested using a Pearson correlation (*r*). 

Further, we sought out the sensibility/specificity of the test (i.e., specific iTBS aftereffects on connectivity patterns) in differentiating patients with MCI at the individual level (i.e., MCI+ and MCI-) by using the likelihood ratio (LR). This analysis provides a summary of how many times more (or less) likely patients with a disease (namely, MCI-) are to have a particular result (in our working hypothesis, specific detrimental connectivity patterns) than patients without the disease (namely, MCI+). In other words, LR can be used to calculate the probability of disease (namely, MCI-) for each individual [95]. 

Last, the receiver operating characteristics (ROC) analysis was used to assess the relative predictive power of specific iTBS-induced modulations in connectivity to predict MDC at six months. We calculated the area under the curves (AUC) and the sensitivities and specificities for MDC using a ROC Analysis Calculator [96,97].

## 3. Results

All patients well tolerated the experimental procedure, reporting no adverse events. Cerebellar iTBS modulated different cortico-cortical connectivity patterns in the majority of patients with MCI, namely MCS+. Specifically, iTBS caused significant variations in resting-state functional connectivity (PLS analysis) (Figure 2), as reflected by the changes in PSD and iCoh between source time series (Figure 3). The post-hoc analyses (EEG phase difference and the Hurst component of iCoh values) revealed details on the dynamics of the connections that were significantly modulated by iTBS (Table 2). On other hand, the sham-iTBS (performed in all enrolled patients) did not yield any significant connectivity aftereffects. 

We had a reduction of functional connectivity, as estimated by multivariate PLS method and the paired *t*-tests to compare post-pre iTBS resting-state connectivity changes, between left somatosensory association cortex (BA5) and left (*p* = 0.002) and right sensorimotor areas (BA2/4) (*p* < 0.001), and between BA5L and left (*p* < 0.001) and right supplementary motor area (BA8) (*p* < 0.001) (Figure 2). On the other hand, there was a strengthening of connectivity between BA5L/R with BA39L (angular) (*p* = 0.001), and left and right BA47 (orbital) (*p* < 0.001, and *p* = 0.002, respectively) (Figure 2). Other connectivity pattern changes were also detectable in patients with MCI+ (including BA47-BA18, BA39-BA08, BA10-BA18, BA47-BA18) but without reaching the statistical significance.

iTBS significantly affected high-alpha and low-beta (11–20 Hz) PSD within BA5L-BA2/4L, BA5L-BA2/4R, and BA5R-BA8R, and the theta and low-alpha (5–8 Hz) PSD within BA5L-BA2/4L and BA5R-BA8R (Figure 3A,B). The PSD and iCoh values were significantly higher following real iTBS in patients with MCI+, as compared to MCI-/dementia (who were grouped together as there were not significant differences between AD, VaD, and MCI-) (Figure 3A,B).

All post-TBS phase spectra increased significantly within the networks and frequency ranges mentioned afore (Table 2). In addition, the temporal evolution of connectivity among the abovementioned areas significantly changed, as estimated by the Hurst exponent (Table 2).

We found a significant negative correlation between overall iTBS aftereffects and the cognitive profile as per Clinical Dementia Rating scale (*r* = −0.496, *p* = 0.001) (Figure 4A). Among patients with MCI, the MCI- showed a clear deterioration of the iTBS-induced changes in BA5-BA8 theta and alpha connectivity, without significant differences between aMCI and naMCI (Figure 4B). We thus sought at intragroup clinical-electrophysiological differences. There were not significant differences between aMCI and naMCI, as well as between AD and VaD. No other correlations between socio-demographic and electrophysiological data were significant.

The relevance of specific detrimental patterns of connectivity within distinct frequency bands was confirmed by LR analysis, which disclosed that the preservation of iTBS-induced changes in BA5-BA8 theta and low-alpha connectivity was of significant utility to differentiate MCI patients at individual level (Figure 5). In fact, we found a LR for a positive result (sensitivity/(1-specificity)) between 5 and 10, which indicates that the test result has a moderate effect on increasing the probability of disease presence (MCI-), and al LR for a negative result ((1-sensitivity)/specificity) between 0.1 and 0.5, which indicates that the test has a moderate effect on decreasing probability of disease (MCI-). The apparent disparity between a moderate effect of 5 to 10 and of 0.1 to 0.5 comes from the different result scale for the LF for positive results (LR+) and the LR for negative results (LR-). The former has values greater than 1, being the higher the score, the more the probability of disease (i.e., more LR); the latter ranges from zero 0 to 1, being the lower the score, the lesser the probability of disease (i.e., less LR). MCI categorization (i.e., aMCI and naMCI) was not relevant to LR analysis. 

All the participants were clinically re-assessed six months after the end of the experimental protocol. All the patients with dementia, MCI+, and three with MCI- (two naMCI and one aMCI) remained substantially stable at the neuropsychological tests, whereas the other five patients with MCI- (four aMCI and one naMCI) converted to dementia. ROC analysis showed that a clear prediction of MDC (AUC = 0.9693; SE = 0.0349) was obtained with the lack of BA5-BA8 theta and alpha connectivity changes after iTBS application (Figure 6). Group categorization (AD vs. VaD, and aMCI vs. naMCI) was not relevant to ROC analysis. 

## 4. Discussion

To the best of our knowledge, this is the first study using PLS investigating the changes in resting state brain-wide functional connectivity after a single session of cerebellar iTBS. We found that the somatosensory association cortex, the sensorimotor areas, and the supplementary motor area showed significant functional connectivity changes after iTBS within theta, alpha, and beta frequency ranges in some patients with MCI (i.e., MCI+). On the contrary, all patients with dementia and the others with MCI (i.e., MCI-) lacked of iTBS aftereffects. There were no significant differences concerning the different subgroups (AD/VaD, a/naMCI). It is noteworthy that no effects were observed after sham-TBS, suggesting that connectivity changes were due to iTBS and not to other, confounding effects. 

As the main finding, cerebellar iTBS shaped the cortico-cortical functional connectivity (with their specific frequency ranges of oscillatory activity) among some of the brain areas that are crucial for sensorimotor integration processes, as indicated by PSD, iCoh, and PLS data analyses [98,99]. Further, the changes in EEG phase difference induced by iTBS suggested that the direction of the effects accompanying the Coh upregulation was from the associative complex to the primary sensorimotor areas. Last, the increase of the Hurst exponent indicated a persistent, long-range temporal dependency of the networks consistent with a slow power-law decay. Altogether, these data further corroborate the current evidences that the cerebellum plays an important role in controlling cortico-cortical connectivity, with particular regard to fronto-parietal, top-down connectivity, as well as in cognitive processes, as suggested by the correlation between iTBS aftereffects and the overall cognitive profile. 

Moreover, cerebellar iTBS aftereffects allowed to differentiate patients with dementia and MCI, and to identify the patients with MCI who may be at risk of MDC. Concerning the first issue, patients with dementia did not show any significant iTBS aftereffects contrary to the patients with MCI. In addition, overall electrophysiological aftereffects were independent from cognitive decline etiology (i.e., AD and VaD), group categorization (i.e., aMCI, and naMCI), the neuroradiological pattern, and the clinical-demographic features of the enrolled patients. This finding may somehow limit the extrapolation of the data to a real-world setting. However, it contemporary confirms the potential feasibility of our electrophysiological approach concerning differential diagnosis at a group level. More interestingly, concerning the second issue, our approach allowed a differential diagnosis at the individual level concerning patients with MCI. In fact, we identified eight patients with MCI who did not properly respond to iTBS (i.e., MCI-, like patients with dementia). Notably, five among these patients converted into dementia at the six-month follow-up, depending on the degree of detrimental BA5-BA8 connectivity in the theta and alpha frequency range, regardless of their subgroup categorization (aMCI vs. naMCI). One may argue that patients with MCI- might be part of a normal variability in the general population, given that some healthy people do not even respond to any particular non-invasive stimulation paradigm. However, the fact that the patients who converted to dementia were those complaining of specific, distinctive connectivity feature, i.e., the failure in BA5-BA8 theta and alpha connectivity, makes unlikely the concern above. Indeed, such a connectivity failure is relevant to several behavioral and cognitive functions (including working memory, motor learning, and different sensorimotor processes) [100,101,102,103,104]. Further, the pathways between such areas can provide the neural substrate for cognitive reserve and compensates for neuropathological changes and dysfunction in other regions owing to its ability to experience neuroplasticity [105,106,107]. Therefore, the role of such connectivity pattern deterioration may be reliable concerning MDC. However, further studies are required to confirm this issue, as well as to clarify whether such bundle represents a potential treatment target in patients with cognitive decline to enhance cognitive function.

To summarize, patients with a clinical diagnosis of MCI lacking of cerebellar neuromodulation-induced changes in cortico-cortical connectivity patterns (as found in patients with dementia) should be strictly followed, as they may be at an increased risk of MDC [108]. However, the relatively small and heterogeneous sample enrolled imposes caution and larger numbers are mandatory to confirm our promising findings. 

### 4.1. Neurophysiological Basis of Cerebellar iTBS Aftereffects

Our study offers also some information regarding the mechanisms of CCC. It is known that the cerebellum can affect the excitability of several cortical areas. Non-invasive neuromodulation can be employed to investigate the contribution of specific cerebellar components and structures to CCC [27,28,29,52,75]. Indeed, cerebellar stimulation can evoke cortical changes in both areas under direct control (including M1 and posterior parietal cortex) [52], and distant areas (including frontal, central and posterior areas bilaterally), probably by targeting specific, local intracortical GABAergic circuits [52,75]. Therefore, it is arguable that the cerebellum contributes to orchestrate connectivity among different cortical areas with potentially important consequences onto diverse physiological processes, including motor learning and memory [45,67,68,109,110].

We extended these findings by demonstrating extensive, top-down changes in brain-wide functional connectivity using PLS (that extracts consistent changes in coherence across all connections). The connectivity dynamics we described agree with previous studies showing that specific areas within the fronto-parietal network (including the associative complex and primary sensorimotor areas) receive cerebellar projections that are necessary to regulate the top-down information flow within the cortex [84,111,112,113]. The specificity of cerebellar iTBS aftereffects is further confirmed by the selective modulation of oscillatory pattern within specific connections. 

Cortical oscillations arise from complex interactions among the thalamus, subcortical areas and the cerebral cortex, namely the thalamo-cortical system. This encompasses both intrinsic (depending on the interactions between specific intrinsic currents) and extrinsic (produced by the interplay of excitatory and inhibitory neurons) oscillatory mechanisms. The GABAergic neurons are attributed to regulate these complex interaction patterns with particular regard to frontoparietal and sensorimotor regions connections [114]. In keeping with the issue that the TBS-induced effects mainly depend on GABAergic interneurons at thalamic and/or cortical level [52,75], we may hypothesize that the TBS mainly acted through GABAergic transmission within cortico-thalamo-cortical system to regulate the oscillatory and connectivity patterns within sensorimotor regions we found.

### 4.2. Limitations

There are some limitations to acknowledge. First, we enrolled a relatively small and heterogeneous sample. This somehow limits the extrapolation of the data to the real world, thus obviously implicating the necessity of larger and more homogeneous samples to confirm our promising findings. Second, we followed-up the patients for a relatively short period, so that the prognostic value of EEG signal analysis needs further confirmation. Third, one may be concerned on the fact that we mapped the electrodes on MNI brain template without taking into account the placement of the electrodes around individual brains. Indeed, the analysis and statistical approaches we used concerning ROI-wise functional connectivity assessment substantially reduces the biasing effect of different sources on the recorded signals, so that it was not mandatory to take into account the placement of the electrodes around individual brains [83,115,116]. It remains anyway necessary to confirm our promising findings employing different analysis approaches.

## 5. Conclusions

Even though the usefulness of the EEG complex analysis we implemented in assessing the clinical progress and estimating prognosis needs to be substantiated more to have relevant implications in real clinical setting, we furnished an objective description of the degree of cognitive dysfunction by estimating connectivity failure. In fact, our study suggests that the degree of cognitive and behavioral impairment is correlated to the deregulation of connectivity among distinct cortical networks that are under cerebellar influence (via phase-lagged interactions within the frontoparietal cortices at theta, alpha, and beta frequency ranges). Such degree of connectivity deregulation supports the differential diagnosis between patients with dementia and MCI and, among the latter, it might identify the patients who may be at risk of MDC, independently from etiology and subgroup categorization. Last, the knowledge of these electrophysiological patterns could be helpful to personalize rehabilitation strategies based on cognitive training and/or non-invasive neuromodulation approaches aimed at restoring large-scale cortico-cortical connections harnessing CCC.

## Figures and Tables

**Figure 1 jcm-08-01470-f001:**
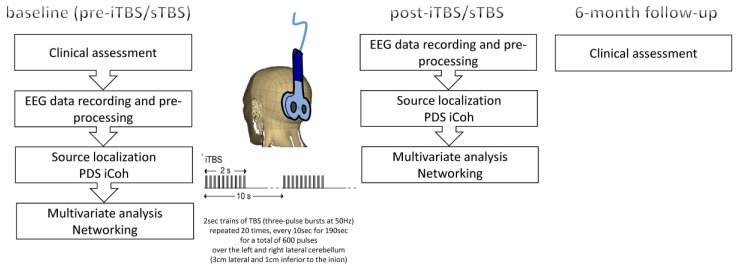
Experimental paradigm. Patients were clinically and electrophysiologically assessed at baseline, after theta-burst stimulation (TBS) delivery, and at the follow-up. iTBS/sTBS-induced post-pre differences in resting-state cortical networks were based on electroencephalographic (EEG) recording and pre-processing, source reconstruction, pair-wise coherence, multivariate partial least square (PLS) analysis of the resulting spectral and coherence data, and resting-state network construction (with spectral content) from PLS data.

**Figure 2 jcm-08-01470-f002:**
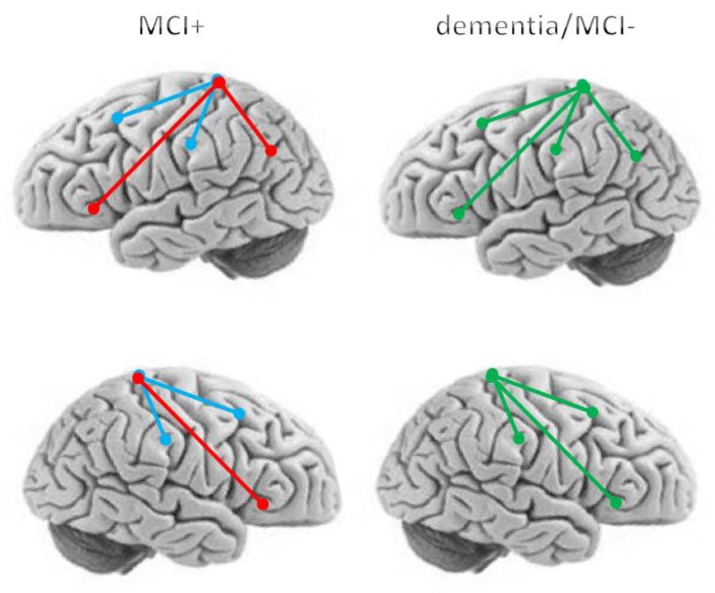
Changes in resting-state functional connectivity from pre- to post-iTBS. The brains plot only the significant changes in cortico-cortical connectivity induced by iTBS within left and right hemispheres (from the top to bottom). Red, blue, and green colors of the spheres (the Euclidean centers of the regions of interests (ROIs)) and lines (voxel connections) indicate the direction (increase, decrease, and no changes, respectively) of connectivity changes as induced by real iTBS. Patients with dementia and Mild Cognitive Impairment (MCI)- showed no changes in connectivity as induced by real iTBS. Subgroups are pooled together as there were not significant differences between VaD and AD, aMCI and naMCI, and AD/VaD and MCI-. Sham iTBS yielded no effects.

**Figure 3 jcm-08-01470-f003:**
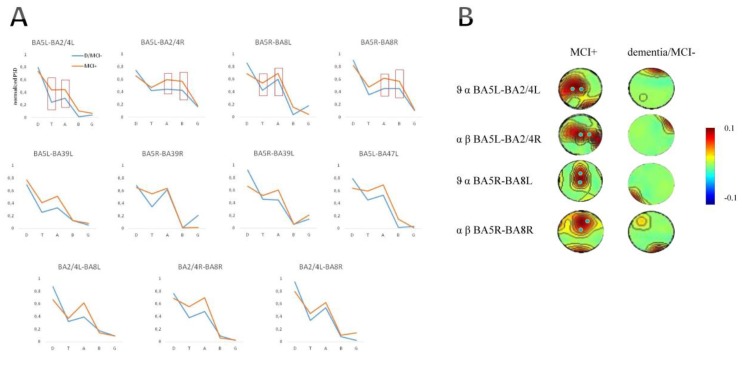
Spectral dynamics of post-pre iTBS resting-state networks. Panel A illustrates the power spectral density (PSD) (normalized by the spectral resolution employed to digitize the signal) of all the networks modified by iTBS, with peaks at the significant carrier frequencies (high-alpha and low-beta frequency range, 11–20 Hz, and theta and low-alpha, 5–8 Hz) (red boxes). Panel B plots the post-pre TBS difference of topographic view of average iCoh values. Post-pre iTBS-induced changes in iCoh are coded in the color-bar (red-increase, blu-decrease). Values are interpolated for areas between electrodes at the significant carrier frequencies in the four significant edges (BA5L-BA2/4L; BA5L-BA2/4R; BA5R-BA8L; BA5R-BA8R).

**Figure 4 jcm-08-01470-f004:**
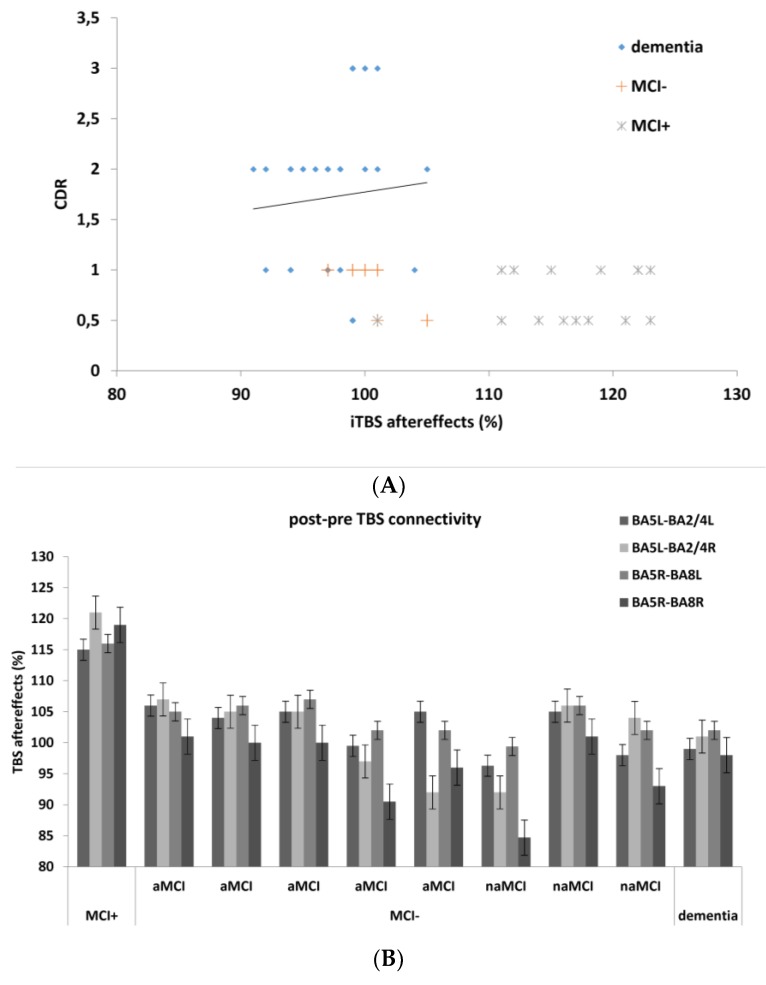
(**A**) shows the scattergram of the correlation between overall cognitive profile (CDR Clinical Dementia Rating scale) and overall intermittent Theta-Burst Stimulation (iTBS) aftereffects (%) in patients with dementia and MCI. (**B**) Among the patients with MCI- (non-responsive to iTBS), aMCI did not significantly differ from the naMCI with regard to the four significant edges and, in particular, to BA5-BA8 theta and low-alpha (5–8 Hz) connectivity.

**Figure 5 jcm-08-01470-f005:**
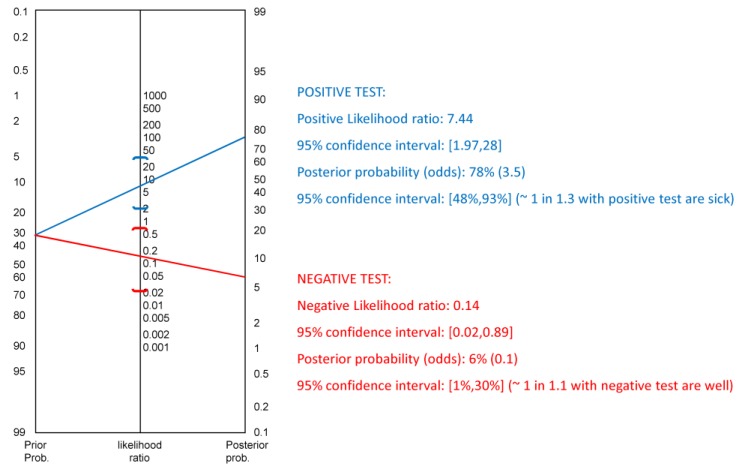
Fagan’s nomogram of the positive and negative LR of the BA5-BA8 theta and alpha connectivity toward MCI differentiation.

**Figure 6 jcm-08-01470-f006:**
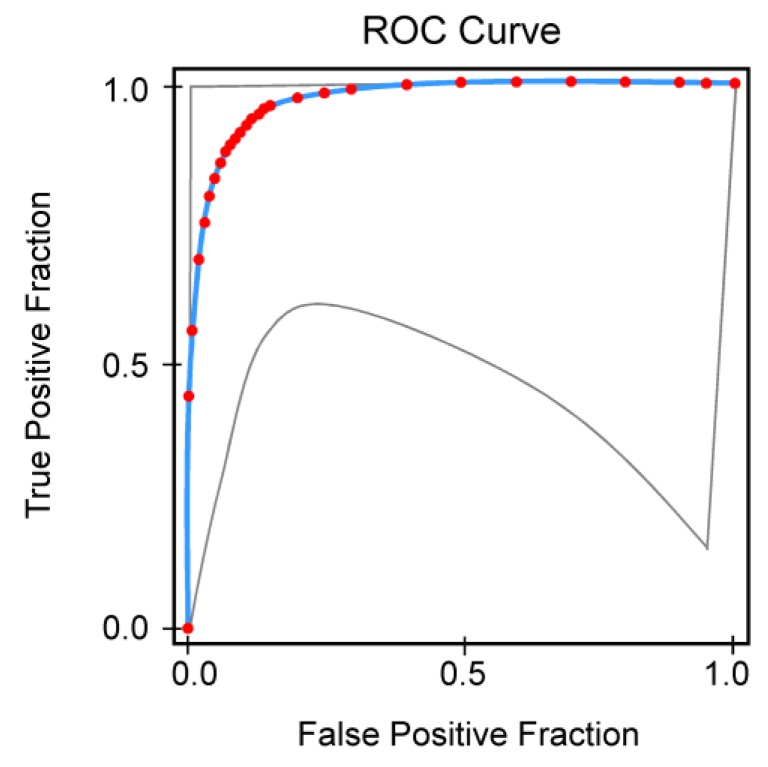
Receiver operating characteristics (ROC) analysis of MCI-to-dementia conversion (MDC) prediction at 6-month follow-up based on the lack of BA5-BA8 theta **and** alpha connectivity changes after iTBS application. Red symbols and blue line indicate fitted ROC curve, gray lines the 95% confidence interval of the fitted ROC curve.

**Table 1 jcm-08-01470-t001:** Clinical-demographic characteristics. Legend: CDR Clinical Dementia Rating scale; dd disease duration; brain volume measure is reported as the percentage of intracranial volume; AD Alzheimer disease; VaD vascular dementia; MCI Mild Cognitive Impairment; an amnestic MCI; na non-amnestic MCI.

Group	Age (y)	Education (y)	dd (y)	CDR	Brain Volume	Medications						
						Donepezil (mg/day)	SSRI	tzD	AED	TA	ATA	Other Drugs
aMCI (*n* = 13)	75	6	4	0.5	0.71		x					x
	75	5	4	0.5	0.74		x					x
	66	11	5	1	0.77						x	
	74	16	3	1	0.8			x				x
	70	9	5	0.5	0.75							
	70	16	2	1	0.75							
	69	11	2	1	0.7		x		x			x
	69	15	3	0.5	0.71		x					x
	68	16	3	0.5	0.74		x					x
	68	17	3	1	0.77						x	
	71	5	3	1	0.71		x					
	67	14	4	1	0.74		x					x
	72	12	2	1	0.76		x					x
mean s.d.	70	12	3	0.7	0.74							
	3	4	1	0.3	0.03							
naMCI (*n* = 12)	69	10	3	1	0.78							
	67	12	5	0.5	0.78		x		x		x	
	74	11	3	0.5	0.79			x				
	65	15	4	1	0.78							
	72	17	5	1	0.79		x					
	71	9	2	0.5	0.76							x
	68	14	4	1	0.76		x					
	69	16	3	1	0.78							
	67	17	4	0.5	0.75							
	65	17	4	1	0.69							
	62	8	5	1	0.71		x		x		x	
	62	11	4	0.5	0.75			x				
mean s.d.	68	13	4	0.8	0.76							
	4	3	1	0.2	0.03							
grand mean s.d.	69	12	4	1	0.75							
	1.9	1	0.3	0.03	0.01							
AD (*n* = 12)	68	16	4	2	0.7	13	x				x	
	67	14	2	1	0.69	18		x	x	x		x
	72	11	4	3	0.72	11		x	x		x	x
	71	7	1	2	0.7	12	x				x	
	71	7	4	2	0.69	13		x	x	x		x
	74	15	3	3	0.6	17	x				x	
	79	9	3	3	0.6	15	x				x	
	75	8	3	3	0.62	15		x	x	x		x
	72	5	4	0.5	0.7	17		x	x	x		x
	77	12	3	1	0.64	17		x	x		x	x
	71	15	1	1	0.61	11	x				x	
	68	13	3	2	0.63	17		x	x	x		x
mean s.d.	72	11	3	2	0.66	15						
	4	4	1	1	0.04	3						
VaD (*n* = 12)	73	17	1	1	0.75	12	x		x	x		x
	73	13	2	2	0.65	13	x				x	
	69	12	4	2	0.77	13				x		x
	71	9	2	2	0.75	13	x		x	x		x
	71	8	1	2	0.65	13	x				x	
	68	17	2	1	0.65	18	x					x
	70	6	4	2	0.67	12	x		x	x		x
	75	8	3	2	0.63	15	x		x	x		x
	74	7	3	2	0.66	15	x				x	
	70	12	2	0.5	0.65	17	x				x	
	70	17	1	0.5	0.63	8				x		x
	77	11	2	1	0.73	12	x		x	x		x
mean s.d.	72	11	2	1	0.68	13						
	3	4	1	1	0.05	3						
grand mean s.d.	72	11	3	2	0.67	14						
	0.2	0.3	0.4	0.3	0.02	1						

**Table 2 jcm-08-01470-t002:** Pre- and post-TBS phase spectra and Hurst exponent values of the four significant edges (BA5L-BA2/4L; BA5L-BA2/4R; BA5R-BA8L; BA5R-BA8R) in patients with dementia (D) and MCI (+/-). W_p within-group post-pre p-value; B_p between-group post-pre p-value.

Pre- and Post-iTBS Phase Spectra (π/2)													
		Pre		Post		W_p	B_p	Pre		Post		W_p	B_p
		theta						alpha					
BA5L-BA2/4L	D/MCI-	0.24	±0.04	0.22	±0.04	0.02	0.009	0.29	±0.04	0.38	±0.03	0.04	0.06
	MCI+	0.33	±0.04	0.35	±0.06	0.005		0.44	±0.06	0.01	±0.01	0.006	
		alpha						beta					
BA5L-BA2/4R	D/MCI-	0.26	±0.05	0.28	±0.05	0.03	0.05	0.26	±0.05	0.45	±0.02	0.003	0.05
	MCI+	0.34	±0.05	0.33	±0.05	0.006		0.32	±0.05	0.02	±0.01	0.008	
		theta						alpha					
BA5R-BA8L	D/MCI-	0.21	±0.04	0.21	±0.04	0.01	0.01	0.28	±0.05	0.63	±0.03	0.37	0.7
	MCI+	0.34	±0.04	0.38	±0.05	0.008		0.41	±0.05	0.01	±0.01	0.006	
		alpha						Beta					
BA5R-BA8R	D/MCI-	0.3	±0.05	0.28	±0.05	0.004	0.03	0.29	±0.05	0.23	±0.03	<0.001	0.02
	MCI+	0.31	±0.05	0.37	±0.06	0.008		0.35	±0.05	0.03	±0.01	0.005	
pre- and post-iTBS Hurst exponent													
		theta						Alpha					
BA5L-BA2/4L	D/MCI-	0.92	±0.16	0.79	±0.16	0.05	0.006	0.3	±0.05	0.02	±0.01	0.004	0.06
	MCI+	0.82	±0.13	0.83	±0.12	0.006		0.28	±0.05	0.02	±0.01	0.005	
		alpha						beta					
BA5L-BA2/4R	D/MCI-	0.3	±0.06	0.31	±0.05	0.02	0.001	0.13	±0.02	0.44	±0.01	0.02	0.07
	MCI+	0.36	±0.05	0.33	±0.06	0.002		0.13	±0.02	0.03	±0.01	0.003	
		theta						alpha					
BA5R-BA8L	D/MCI-	0.83	±0.17	0.99	±0.17	0.03	0.006	0.34	±0.06	0.07	±0.17	0.002	0.00
	MCI+	0.93	±0.15	0.88	±0.14	<0.001		0.41	±0.05	0.02	±0.01	0.006	
		alpha						beta					
BA5R-BA8R	D/MCI-	0.31	±0.05	0.28	±0.05	0.02	0.03	0.13	±0.02	0.4	±0.01	0.01	0.50
	MCI+	0.28	±0.04	0.33	±0.05	0.01		0.12	±0.02	0.04	±0.01	0.002	

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
