# Peer review of "New Horizons in Early Dementia Diagnosis: Can Cerebellar Stimulation Untangle the Knot?"

_jcm, 2019, doi:10.3390/jcm8091470_

Round 1
Reviewer 1 Report
The manuscript has been substantially improved since the original submission. One of the main improves concerns the increased number of participants, which now is able to provide some meaningful results for the community. However, there are still some issues which require more details.
Major issues:
1. The authors classify the dementia patients in AD and VaD. However, they do not indicate any bio-marker for the detection of AD. According to current consensus, this patients can not be diagnosed as AD without bio-markers. Please, modify the assertion, for example using old NINCDS-ADRDA "probable AD" classification.
2. Some of the new MCI patients have (normalized) brain volumes as low as 0.27, 0.23 or 0.15. These brain (white plus grey matter) volumes do not make sense. I suggest the authors to review them.
3a. The "Experimental procedure" (page 5, lines 119 to 126) and the "TMS procedure" sections overlap. I suggest to rephrase both sections or even merge them.
3b. The "Experimental procedure" section includes a justification for the use of iTBS (page 5, lines 135 to 140). This justification should appear in the introduction, not in the methods section.
3c. The selection of the threshold for the TMS (section "TMS procedure") is fragmented (one part in page 5, lines 144 to 148, other part in page 6, 156 to 159). In order to clarify the exposition, I would consider rephrasing the section, writing first the definition of the TMS system, then the procedure for the determination of the threshold, and last the experimental TMS application.
4. In the "EEG recording and analysis" section, the authors indicate that the per-ROI time series was extracted by averaging the time series of all the sources in he ROI (page 6, lines 186 to 187) and by taking the first PCA of those sources (same page, lines 195 to 196). Which one of both procedures was used?
5 The authors continue talking about tfCoh, when (and maybe I am wrong) the temporal component is never used. Then, in the results section (page 7, lines 215 to 216), the authors mention they use imaginary coherence, but do not introduce it in the methods section. Was the imaginary part of coherence used on all the statistical comparisons? It is not clear if it was only used on some of them.
6 In page 7, line 223, the authors mention that they established the significance of the coherence values using a set of 1000 permutations. This number is extremely low to attain a good confidence level in the estimated p-value.
7 In page 8, lines 257 to 263, the authors claim to have found differences between MCI+ and MCI- patients using the iTBS after-effects measured in connectivity. As MCI+ (and MCI-) are defined as those patients which do (or do not) present iTBS after-effects, this comparison is circular, and should be removed.
8 Figure 2 is extremely confusing. What are the circulograms supposed to mean? Whay are Brodmann areas related to single electrodes in the circulogram? In addition, the colors of the links should indicate the change (increase/no change/decrease), as the think lines are difficult to see, yet important.
9 In page 8, lines 281 to 283, the authors report values for PSD. I suppose this means power spectral density, but this metric was not included in the methods section, and it is not clear how it is calculated. Please, include all the methodological procedures in the methods section.
10 In Figure 4, part A, both graphs show the same points. These two graphs should be merged, for example taking the bottom one and marking MCI+ with a colored asterisk and MCI- with a colored plus circle, in order to distinguish them. In addition, the value for the after-effects in one aMCI- patient changes between both graphs (CDR = 0.5, iTBS after-effects ~ 105 in the upper graph, 109 in the lower graph).
11 In page 10, lines 321 to 325, a LR of between 5 and 10 is marked as moderate effect, while one between 0.1 and 0.5 is also marked as moderate. I am not familiar with likelihood ratios, but this disparity sounds strange.
12 In page 11, line 335, the authors say they use the theta-alpha connectivity to generate a classifier. What do they refer to with theta-alpha connectivity? This phrasing is usually employed for cross-frequency coupling, but the authors do not use this connectivity method in the manuscript.
13 Along the manuscript, the authors refer to the Hurst exponent. This method is not detailed in the methods section, and its significance is not stated, either.
14 In page 13, lines 431 to 433, the authors say that the volume conduction could be affecting the results of imaginary coherence. Imaginary coherence is insensitive to volume conduction, so this does not make sense.
Minor issues:
1. The authors use the phrasing "cerebellum-cerebrum connectivity". I suggest them to change it to an acronym, for example with the initials "CCC" or "C-C connectivity".
2. With the new participants, in page 3, line 104, the authors claim that the participants were enrolled "between March and January 2019". I suppose they refer to March 2018.
3. In page 6, line 177, the author use the phrasing "Montreal Neurological Institute (MNI)152 template". The name of the template is MNI152 or ICBM152, so I would change it to "Montreal Neurological Institute template (MNI152)".
4 In page 6, lines 197 to 200, the authors mentioned twice that the application of iTBS is a generic intervention. It looks like the authors forgot to remove one of the assertions. In its current form, the phrasing is reiterative.
5. In page 7, lines 242-243, the authors write a link to the ROC calculator used for the analysis. I suggest this link is moved to a footnote or, if not allowed by the journal, a reference.
6. In page 12, lines 376 to 377, the authors claim that up to a third of participants show no iTBS after-effects. The number of MCI patients in the study is 25, and 8 is exactly one third of 25. So this phrase opens a lot of questions. I would consider reviewing it.
Author Response
We thank the reviewer for the appreciation for our manuscript and for the further useful comments to improve its quality.
The authors classify the dementia patients in AD and VaD. However, they do not indicate any bio-marker for the detection of AD. According to current consensus, this patients can not be diagnosed as AD without bio-markers. Please, modify the assertion, for example using old NINCDS-ADRDA "probable AD" classification.
Checked and corrected.
Some of the new MCI patients have (normalized) brain volumes as low as 0.27, 0.23 or 0.15. These brain (white plus grey matter) volumes do not make sense. I suggest the authors to review them.
Checked and corrected.
The "Experimental procedure" (page 5, lines 119 to 126) and the "TMS procedure" sections overlap. I suggest to rephrase both sections or even merge them.
Accordingly the two sections were merged.
The "Experimental procedure" section includes a justification for the use of iTBS (page 5, lines 135 to 140). This justification should appear in the introduction, not in the methods section.
This part was moved to the introduction, as suggested.
The selection of the threshold for the TMS (section "TMS procedure") is fragmented (one part in page 5, lines 144 to 148, other part in page 6, 156 to 159). In order to clarify the exposition, I would consider rephrasing the section, writing first the definition of the TMS system, then the procedure for the determination of the threshold, and last the experimental TMS application.
The “TMS” section, now merged with the “experimental procedure” section, was rephrased as suggested.
In the "EEG recording and analysis" section, the authors indicate that the per-ROI time series was extracted by averaging the time series of all the sources in he ROI (page 6, lines 186 to 187) and by taking the first PCA of those sources (same page, lines 195 to 196). Which one of both procedures was used?
We rewrote this paragraph since misleading. We have now better specified that we first identified the centroid voxel within each ROI to extract the orientation of the source-dipole dynamics (source-reconstructed EEG); then, we applied principal component analysis (PCA) to each of the centroid voxels separately. The first PCA projection of each centroid voxel time series was considered for the data analysis of source-reconstructed EEG.
The authors continue talking about tfCoh, when (and maybe I am wrong) the temporal component is never used.
In keeping with reviewer concern, we opted to avoid such confounding term (time dimension collapse vs. post-pre differences in resting state EEG) and thus simply referred to Coh, which was more detailed, as well.
Then, in the results section (page 7, lines 215 to 216), the authors mention they use imaginary coherence, but do not introduce it in the methods section. Was the imaginary part of coherence used on all the statistical comparisons? It is not clear if it was only used on some of them.
The description of Coh was entirely revised and added of the missing information; the iCoh was introduced as well. The iCoh was used on all the statistical comparisons, as now better specified.
In page 7, line 223, the authors mention that they established the significance of the coherence values using a set of 1000 permutations. This number is extremely low to attain a good confidence level in the estimated p-value.
Corrected. We apologize for this mere typo error.
In page 8, lines 257 to 263, the authors claim to have found differences between MCI+ and MCI- patients using the iTBS after-effects measured in connectivity. As MCI+ (and MCI-) are defined as those patients which do (or do not) present iTBS after-effects, this comparison is circular, and should be removed.
These comparison were removed, as suggested.
Figure 2 is extremely confusing. What are the circulograms supposed to mean? Whay are Brodmann areas related to single electrodes in the circulogram? In addition, the colors of the links should indicate the change (increase/no change/decrease), as the think lines are difficult to see, yet important.
In keeping with reviewer’s concern and the objective difficulty in interpreting Fig. 2A, this panel was deleted. Fig.2b was redrawn as suggested. Now, colors indicate the direction of iTBS-induced connectivity change, whereas the continuity of the lines indicates the hemisphere to which they belong.
In page 8, lines 281 to 283, the authors report values for PSD. I suppose this means power spectral density, but this metric was not included in the methods section, and it is not clear how it is calculated. Please, include all the methodological procedures in the methods section.
We added the missing information in the method section.
In Figure 4, part A, both graphs show the same points. These two graphs should be merged, for example taking the bottom one and marking MCI+ with a colored asterisk and MCI- with a colored plus circle, in order to distinguish them. In addition, the value for the after-effects in one aMCI- patient changes between both graphs (CDR = 0.5, iTBS after-effects ~ 105 in the upper graph, 109 in the lower graph).
The panel A of Fig.4 was modified as usefully suggested by the reviewer. Patients’ values were checked, as well.
In page 10, lines 321 to 325, a LR of between 5 and 10 is marked as moderate effect, while one between 0.1 and 0.5 is also marked as moderate. I am not familiar with likelihood ratios, but this disparity sounds strange.
The disparity is indeed only apparent, as it comes from the different result scale for the likelihood ratio for positive results (LR+) and the likelihood ratio for negative results (LR). The former has values greater than 1, being the higher the score, the more the probability of disease (+LR); the latter ranges from zero 0 to 1, being the lower the score, the lesser the probability of disease (-LR). So, both the LR values can be marked as moderate effect. We added this comment in the result section.
In page 11, line 335, the authors say they use the theta-alpha connectivity to generate a classifier. What do they refer to with theta-alpha connectivity? This phrasing is usually employed for cross-frequency coupling, but the authors do not use this connectivity method in the manuscript.
The reviewer is right. The phrasing (that has been revised) is indeed misleading as we were not referring to cross-frequency coupling, but simply to theta and alpha frequency range connectivity.
Along the manuscript, the authors refer to the Hurst exponent. This method is not detailed in the methods section, and its significance is not stated, either.
We introduced and detailed the concept of the Hurst exponent in the method section.
In page 13, lines 431 to 433, the authors say that the volume conduction could be affecting the results of imaginary coherence. Imaginary coherence is insensitive to volume conduction, so this does not make sense.
This misleading period, as correctly pointed out by the reviewer, was deleted.
Minor issues:
The authors use the phrasing "cerebellum-cerebrum connectivity". I suggest them to change it to an acronym, for example with the initials "CCC" or "C-C connectivity".
Done.
With the new participants, in page 3, line 104, the authors claim that the participants were enrolled "between March and January 2019". I suppose they refer to March 2018.
Right, corrected.
In page 6, line 177, the author use the phrasing "Montreal Neurological Institute (MNI)152 template". The name of the template is MNI152 or ICBM152, so I would change it to "Montreal Neurological Institute template (MNI152)".
Done.
In page 6, lines 197 to 200, the authors mentioned twice that the application of iTBS is a generic intervention. It looks like the authors forgot to remove one of the assertions. In its current form, the phrasing is reiterative.
Checked and corrected.
In page 7, lines 242-243, the authors write a link to the ROC calculator used for the analysis. I suggest this link is moved to a footnote or, if not allowed by the journal, a reference.
Done.
In page 12, lines 376 to 377, the authors claim that up to a third of participants show no iTBS after-effects. The number of MCI patients in the study is 25, and 8 is exactly one third of 25. So this phrase opens a lot of questions. I would consider reviewing it.
The sentence was revised accordingly.
Kindest regards,
The authors
Reviewer 2 Report
Authors performed a good revision according to my indications.
Author Response
We thank the reviewer for the appreciation for our manuscript.
Kindest regards,
The authors
Round 2
Reviewer 1 Report
This version of the manuscript show improvement over the last one, but some issues remain.
First, and of paramount importance, the table with the socio-demographic data still shows some extremely weird values. Six of the AD patients has ages below 50, one of the them shows an age of 21, which makes no sense. In the same way, six of the VaD patients show also ages between 50, with the younger being 20 years old. Some of these patients show extremely low brain volumes, with a 27% of the cranial volume filled with real brain in one case.I am quite confident those values are errors, but that kind of errors in a table, where the only required task is to copy values, rises important concerns about the rest of the work.
Minor issues:
In page 5, line 158, the authors talk about "stimulated sides" of the cerebellum. I would use the word "hemispheres" instead of "sides".
In page 6, lines 191 to 195, the method used to extract the representative time series is still confusing. I would consider rephrasing.
In page 7, line 210, the authors refer tot he magnitude coherence as "real-valued". This can cause confusion, due to the opposition with the imaginary coherence described below. I would use the name "magnitude coherence" or just "coherence" (as the complex value is termed coherency).
In page 7, lines 216 to 222, the authors say that they correct the imaginary part of coherency with the real counterpart. However, they refer to this real counterpart as magnitude-squared coherence, what is wrong (see previous note). The correct naming is: 1) coherence or magnitude coherence (squared or not) for the absolute value; imaginary part of coherency (sometimes imaginary part of coherency) to the imaginary part, real part of coherency (some times real part of coherence) to the real part. In addition, the imaginary part of coherency, corrected by the real counterpart, is sometimes termed corrected imaginary part of coherency/coherence. Please, use the right terms to refer to each values.
Overall, in page 7, lines 210 to 222, the phrasing is confuse, with very long phrases. I would consider rewriting this part, so the procedure is more clear to the reader.
In page 7, line 248, the authors use the word "esteem" when I think they should use "estimate".
In page 8, Figure 2, the authors indicate that a) solid lines refer to right hemisphere, dashed lines refer to left hemisphere and dotted lines refer to inter-hemispheric connections. However, some dashed and dotted lines overlap with solid lines. Is this maybe due to showing only one view on the brain? I find this choice confusing.
Author Response
Dear Editor,
We want to thank you and your reviewers for the appreciation for our manuscript and for the useful comments to improve its quality.
To the reviewer#1:
We thank the reviewer for the appreciation for our manuscript and for the further useful comments to improve its quality.
First, and of paramount importance, the table with the socio-demographic data still shows some extremely weird values. Six of the AD patients has ages below 50, one of the them shows an age of 21, which makes no sense. In the same way, six of the VaD patients show also ages between 50, with the younger being 20 years old. Some of these patients show extremely low brain volumes, with a 27% of the cranial volume filled with real brain in one case. I am quite confident those values are errors, but that kind of errors in a table, where the only required task is to copy values, rises important concerns about the rest of the work.
We thank the reviewer for having underlined these typo errors, which we missed in the former revision rounds. We carefully revised the clinical demographic data of the patients, and consequently checked the related correlation analysis (whose data were confirmed); we found only typo errors in age reporting. Concerning the brain volumes data the task is not copy-paste, as we had to convert the data brain volume manually to fill the table. We however apologize for these errors and we want to thank once again the reviewer for his accurate revision, which further improved the quality of our work.
Minor issues:
In page 5, line 158, the authors talk about "stimulated sides" of the cerebellum. I would use the word "hemispheres" instead of "sides".Corrected.
In page 6, lines 191 to 195, the method used to extract the representative time series is still confusing. I would consider rephrasing.The paragraph was revised and restructured, so to make it more readable.
In page 7, line 210, the authors refer to the magnitude coherence as "real-valued". This can cause confusion, due to the opposition with the imaginary coherence described below. I would use the name "magnitude coherence" or just "coherence" (as the complex value is termed coherency).Corrected.
In page 7, lines 216 to 222, the authors say that they correct the imaginary part of coherency with the real counterpart. However, they refer to this real counterpart as magnitude-squared coherence, what is wrong (see previous note). The correct naming is: 1) coherence or magnitude coherence (squared or not) for the absolute value; imaginary part of coherency (sometimes imaginary part of coherency) to the imaginary part, real part of coherency (some times real part of coherence) to the real part. In addition, the imaginary part of coherency, corrected by the real counterpart, is sometimes termed corrected imaginary part of coherency/coherence. Please, use the right terms to refer to each values.Checked and corrected according to reviewer’s indications.
Overall, in page 7, lines 210 to 222, the phrasing is confuse, with very long phrases. I would consider rewriting this part, so the procedure is more clear to the reader.The paragraph was revised and restructured, so to make it more readable.
In page 7, line 248, the authors use the word "esteem" when I think they should use "estimate".Corrected.
In page 8, Figure 2, the authors indicate that a) solid lines refer to right hemisphere, dashed lines refer to left hemisphere and dotted lines refer to inter-hemispheric connections. However, some dashed and dotted lines overlap with solid lines. Is this maybe due to showing only one view on the brain? I find this choice confusing.Fig.2 was redrawn as simplified, thus showing both hemispheres, as suggested by the reviewer.
Kindest regards,
The authors
This manuscript is a resubmission of an earlier submission. The following is a list of the peer review reports and author responses from that submission.
Round 1
Reviewer 1 Report
The work by Naro et al is amed at improving the differential diagnosis between dementia ad MCI, as well as identifying those patients with MCI at risk of converting into dementia.
The study is well conducted, and results convincing. However, i have some minor concerns to further improve the paper
- although well written, some periods concerning brain connectivity in the introduction can be difficult to follow; please clarify some concepts for readers who are not confident with the issue.
- A figure concerning TMS procedure should be added to help the readers to better understand the methodology.
Reviewer 2 Report
The authors study the affect of cerebellar TMS in the cortical functional connectivity of patients with MCI and dementia. They find that most of the MCI patients show an after-effect related to the stimulation, and none of the dementia patients (plus six MCI patients) show no response. The authors propose this response to TMS as a biomarker for classification and prognosis of evolution to dementia.
While the issue addressed is of great importance, I find several limitations that, in my opinion, negatively affect the impact of the results. Those are listed below.
* An important flaw in the design is the sample size and heterogeneity. MCI and dementia are known for its heterogeneity, and the originally correct (but still small) sample size (19 MCI and 14 dementia patients) is drastically reduced when dividing the data in subgroups (8 non-amnesic MCI, 11 amnesic MCI, 6 vascular dementia, 5 Alzheimer's dementia and 3 mixed dementia patients). This heterogeneity completely avoids for any extrapolation of the data to the real world. At least 12 to 15 participants per subgroup should be used to get some confidence on the results.
* An important flair in the analysis is related to the source reconstruction. In page 5, line 166-168 the authors indicate that the forward model is based on a canonical template based on MNI. Then, in page 5, line 174, the authors indicate that they segment the canonical brain according to the AAL atlas.
** First, the use of a canonical template and canonical electrode positions for the calculation of the forward model for connectivity in brain is usually a bad idea, specially with a small sample. See for example Human Brain Mapping 2018 39:104–119.
** Second, the AAL has 45 brain areas per hemisphere, of which approximately 38 can be considered cortical. This is 76 cortical areas between both hemispheres, and the representative activity of the area is calculated using the PCA of all the dipoles in the area. However, the activity is reconstructed from 19 electrodes. It is impossible to reconstruct the activity for 76 independent areas from 19 electrodes. This implies that the correlation (and, by extension, coherence) of the signals is going to be very high, especially for nearby areas. For example, the high connectivity found between BA5 and BA6 is quite likely due to leakage.
* In the line of previous issue, the authors said they used the AAL atlas to make partitions in the brain, but the results are reported using a Brodmann atlas. Can they explain?
* The authors use time-frequency coherence for the analysis of resting-state data. This does not make sense, and in fact they collapse the time dimension to get the results, as in resting time has not meaning across epochs. All references to tfCoh should be removed, as the authors are really not using a time-frequency extension, but the raw coherence.
* The description of the statistical test used in this work is confuse. In addition, the prognosis analysis showed in Figure 4 is not clear, as the ROC curve has 9 points (when there are 19 MCI patients) and it is not clear how the accuracy/sensitivity/specificity were calculated.
In addition, I found some minor issues:
* The use of TBS should be indicated in the abstract and the introduction. It is not clear, until page 4, that iTBS is executed through TMS.
* In page 2, lines 49-50, the authors talk about "spectral coherence and synchronization". As coherence is a synchronization metric, this phrasing does not make sense.
* In Table 1 the sociodemographic data for each participant is listed. One of the entries is percentage of females. Could you explain the meaning of this entry?
* In page 4, line 140, the shape and manufacturer of the coil is repeated after same page lines 134-136. This is reiterative.
* In page 5, line 156, electrodes are said to be "displaced according to 10-20 International System". The word displaced is wrong, as the electrodes are placed according to this system.
* In page 4 line 158 the authors indicate that they use a EOG to capture blinks and eye movements. Was it a vertical EOG?
* In page 4 line 160 the authors indicate that the impedance was always below 5 KOhms. The authors should replace the lower-or-equal sign by a phrasing, as these kind of symbols should not be used in written paragraphs.
* In page 4 line 165 the authors refer to sLORETA as "Standard low resolution electrical tomography". sLORETA stands for standardized low resolution electrical tomography. Please correct.
* In page 6, lines 198-201, the authors describe the surrogate design after the description of the significant threshold. The order should be inverted, and the method used to construct the surrogates should be explained before explaining how to use them.
* In page 6, lines 215 to 223, the authors claim that some of the links showed significantly higher or lower connectivity values than the rest. How was the significance determined?
* In page 7, line 242, the authors refer to the connections that are strengthen after iTBS as "deficient". There is not reason to call these connections as deficient, as the "normal" range of the connection is not known (as there is no control group).
* In page 9, lines 262-263, the authors say that "Sham-TMS was performed in all the enrolled patients. We did not identify any significant TBS aftereffects". This should be indicated at the beginning of the results section, as the validity of other results lie on this one.
* In page 10, lines 272-273, the authors claim that "The six patients with MCI- behaved as outliers, showing no substantial TBS aftereffects". This is a circular argument, as those patients were identified as MCI- because of the lack of response to the stimulation.
* In page 11, line 305, the authors say that the sTBS showed connectivity changes in "nearly all the patients with MCI". 13 of the 19 original MCI patients are 68%, or 2 out of three. The effect is consisten, but the use of "nearly all" is not justified at all.
* In page 12, lineas 343-345, the issue with the low sample is indicated. This was already indicated with almost an exact phrasing in same page, lines 322-323, and is thus reiterative.
* In In page 12 line 350 the authors mention "cerebellum-brain connectivity". The cerebellum is part of the brain. The authors are likely referring to cerebellum-cerebrum connectivity.